# Research on Soil Moisture Estimation of Multiple-Track-GNSS Dual-Frequency Combination Observations Considering the Detection and Correction of Phase Outliers

**DOI:** 10.3390/s23187944

**Published:** 2023-09-17

**Authors:** Xudong Zhang, Chao Ren, Yueji Liang, Jieyu Liang, Anchao Yin, Zhenkui Wei

**Affiliations:** College of Geomatics and Geoinformation, Guilin University of Technology, Guilin 541004, China; 1020211828@glut.edu.cn (X.Z.); lyjayq@glut.edu.cn (Y.L.); liangjieyu@glut.edu.cn (J.L.); yinanchao@glut.edu.cn (A.Y.); weizhenkui@glut.edu.cn (Z.W.)

**Keywords:** GNSS-IR, soil moisture, dual-frequency pseudorange combination, dual-frequency carrier phase combination, detection and correction of outliers, multipath error

## Abstract

Soil moisture (SM), as one of the crucial environmental factors, has traditionally been estimated using global navigation satellite system interferometric reflectometry (GNSS-IR) microwave remote sensing technology. This approach relies on the signal-to-noise ratio (SNR) reflection component, and its accuracy hinges on the successful separation of the reflection component from the direct component. In contrast, the presence of carrier phase and pseudorange multipath errors enables soil moisture retrieval without the requirement for separating the direct component of the signal. To acquire high-quality combined multipath errors and diversify GNSS-IR data sources, this study establishes the dual-frequency pseudorange combination (DFPC) and dual-frequency carrier phase combination (L4) that exclude geometrical factors, ionospheric delay, and tropospheric delay. Simultaneously, we propose two methods for estimating soil moisture: the DFPC method and the L4 method. Initially, the equal-weight least squares method is employed to calculate the initial delay phase. Subsequently, anomalous delay phases are detected and corrected through a combination of the minimum covariance determinant robust estimation (MCD) and the moving average filter (MAF). Finally, we utilize the multivariate linear regression (MLR) and extreme learning machine (ELM) to construct multi-satellite linear regression models (MSLRs) and multi-satellite nonlinear regression models (MSNRs) for soil moisture prediction, and compare the accuracy of each model. To validate the feasibility of these methods, data from site P031 of the Plate Boundary Observatory (PBO) H_2_O project are utilized. Experimental results demonstrate that combining MCD and MAF can effectively detect and correct outliers, yielding single-satellite delay phase sequences with a high quality. This improvement contributes to varying degrees of enhanced correlation between the single-satellite delay phase and soil moisture. When fusing the corrected delay phases from multiple satellite orbits using the DFPC method for soil moisture estimation, the correlations between the true soil moisture values and the predicted values obtained through MLR and ELM reach 0.81 and 0.88, respectively, while the correlations of the L4 method can reach 0.84 and 0.90, respectively. These findings indicate a substantial achievement in high-precision soil moisture estimation within a small satellite-elevation angle range.

## 1. Introduction

Soil moisture (SM) constitutes a pivotal variable in terrestrial water and energy cycles, with significant implications for fields such as meteorology, agriculture, and drought and landslide prediction [1,2,3,4,5]. However, the traditional processes employed to measure soil moisture are intricate, time consuming, labor intensive, and inadequate for acquiring soil moisture information over large areas. In contrast, remote sensing techniques offer an efficient approach for obtaining soil moisture information. Microwave remote sensing has witnessed notable advancements through the deployment of satellites such as NASA’s Soil Moisture Active Passive (SMAP) and ESA’s Soil Moisture and Ocean Salinity (SMOS), which provide global soil moisture products covering extensive geographical areas [6,7]. Additionally, optical remote sensing has made progress with the launch of Sentinel satellites by the European Space Agency (ESA) and Landsat satellites by NASA/USGS, facilitating the extraction of soil moisture information from optical images [8,9]. The afore-mentioned remote sensing products are suitable for application in vast regions but are not applicable to smaller areas.

In addition to the fundamental positioning, navigation, and timing (PNT) functions of the global navigation satellite system (GNSS) satellites [10], GNSS radio occultation (GNSS-RO) has emerged as a remote sensing application of GNSS satellites, which enables high-resolution estimation of total electron content (TEC) and the spatial distribution of tropospheric water vapor content [11,12]. Simultaneously, the presence of multipath effects arising from satellite signal reflection on the surface of objects represents an inevitable error source in satellite navigation and positioning. However, these errors can be leveraged for estimating surface physical parameters, leading to the development of GNSS reflectometry (GNSS-R) technology, which encompasses GNSS interferometric reflectometry (GNSS-IR) [13].

The GNSS-R technique relies on the left and right antennas of a dual-antenna receiver to receive direct and reflected signals from satellites, respectively, and uses the reflected signals to monitor information on parameters such as sea surface wind fields [14], sea level [15], sea ice thickness [16], soil moisture [17,18], vegetation height [19], and snow depth [20]. However, the application of this technology is limited by the cost and hardware requirements of customized receivers. GNSS-IR technology, meanwhile, estimates surface and ocean parameters, such as soil moisture [21,22,23], vegetation water content [24], snow depth [25,26], sea surface elevation [27], and sea ice [28], through the interfering signals formed by the superposition of the direct and reflected signals at the center of the antenna of a single-antenna receiver. The maximum distance from the center of the antenna is about 45 m at an altitude angle of 5°. The specular reflection point of the satellite signal on the surface of the medium as well as the effective monitoring area are fixed [20]. The signal-to-noise ratio data of different systems can be used to retrieve the soil moisture, and the soil moisture estimated by the GPS L1 and L2P band signals is closer to the true value of soil moisture than the L2C band [29]. The B1 and B2 band signals of the BeiDou satellite navigation system both respond well to the fluctuation of surface soil moisture [30]. In order to unite the signal-to-noise ratio data from more satellites to analyze the soil moisture changes, Liang fused multiple satellites to retrieve soil moisture using least squares support vector machine, and the soil moisture retrieval accuracy of the multi-satellite fusion was significantly improved compared with that of a single satellite [31].

Presently, the majority of scholarly research in the field of GNSS-IR predominantly centers around the processing and analysis of the signal-to-noise ratio (SNR). However, for most users, the SNR value has limited practicality and is not consistently regarded as a fixed observation within GNSS observation files, especially when early SNR data are not consistently recorded in the raw observation files [32]. In order to enhance the versatility of GNSS Interferometric Reflectometry (GNSS-IR) technology, it becomes imperative to explore alternatives to SNR. Assuming continuous satellite signal lock, nearly all GNSS receivers possess the capability to process single-frequency signals and persistently record carrier phase and pseudorange measurements. Consequently, more and more scholarly literature has focused on the estimation of surface physical parameters by utilizing multipath errors of combined observation signals.

In 2012, Ozeki et al. were the pioneers in proposing and applying the L4 combination for snow depth measurement. This method, involving the linear combination of carrier phase observations via inter-frequency differencing, is independent of geometric parameters such as receiver and satellite coordinates. However, following the differencing process designed to negate tropospheric delay, it is noteworthy that the combined ionospheric delay still remains. The effectiveness of this combination in estimating snow depth largely depends on whether higher-order polynomials or a low-pass filter can adequately address the ionospheric delay [32]. Subsequently, in 2015, Yu et al. introduced the idea of combining GPS three-frequency carrier phase observations for snow depth estimation. This approach, when contrasted with the L4 combination, successfully eradicated ionospheric delay and generated high-precision estimates of snow depth [13]. In the following years, Yu et al. in 2018, followed by Li et al. in 2019, introduced the concepts of single-frequency and dual-frequency carrier phase and pseudorange combinations, respectively. Both combination methods exhibit independence from geometric factors, ionospheric delay, and tropospheric delay [33,34]. It is also noteworthy that combined GNSS observations can be effectively utilized for inversion of sea surface height [35,36,37]. However, in comparison to the snow surfaces and sea levels, the unique reflection characteristics of soil surfaces introduce variations in the soil moisture estimation model. This distinction partly explains the relatively delayed adoption of combined GNSS observations for soil moisture estimation. Some researchers have adopted L-S spectral analysis to select satellites with high-quality multipath errors during the observational period. Through equal-weighted fusion of multiple satellite orbital parameters, they have realized the multi-satellite joint inversion of soil moisture. In 2021, Zhang et al. pioneered the estimation of soil moisture using GPS triple-frequency combination observations. They employed triple-frequency carrier phase combination (TRFCP) and triple-frequency pseudorange combination (TRFP) to quantify multipath errors. The results showed that delay phases exhibited a strong correlation with in situ soil moisture at the measurement station [38]. Owing to the limited availability of satellites capable of transmitting three-frequency signals, Nie et al. proposed the amalgamation of dual-frequency GNSS observations in 2022 to enhance the temporal resolution in soil moisture estimation. Rigorous validation using data from nearby stations was conducted to evaluate the efficacy of this approach. The findings revealed that, when compared to the dual-frequency pseudorange combination method (DFP), the dual-frequency carrier phase combination method (L4) displayed superior accuracy in soil moisture estimation [39]. After screening satellites using L-S spectral analysis, satellites with low-quality combined multipath errors are eliminated. When estimating soil moisture at a station by combining the delayed phases of multiple satellite orbits, the reduction in the number of available satellites results in delayed phases of some satellite orbits cannot be used to estimate soil moisture at the station. At the same time, due to the non-robustness of the least squares method, outliers are introduced when the delayed phases are settled by low-quality combined multipath errors. In order to obtain high-quality delayed phases and thus utilize the characteristic parameters of more satellite orbits to estimate soil moisture, outliers need to be detected and corrected. Liang et al. utilized the interquartile range and moving average filter to realize the detection and repair of anomalous phases, and they introduced the multiple linear regression model (MLR) to fuse the multi-star phase inversion of soil moisture, and the inversion of soil moisture in different time periods, all of which obtained high prediction accuracies [40].

This study introduces an innovative approach for estimating soil moisture using GNSS dual-frequency combination observations, with a particular focus on the detection and correction of abnormal phases and the influence of environmental factors on the predictions of different models. It is substantiated that the dual-frequency pseudorange combination (DFPC) and dual-frequency carrier phase combination (L4), which eliminate the influence of tropospheric delay and geometric distance factors, hold potential for accurate soil moisture estimation. Combined multipath errors for five epochs are selected from the optimal elevation range of each satellite, and we construct the error equation to solve the delay phase. The occurrence of outliers in the delay phase is attributed to the suboptimal quality of dual-frequency combined multipath errors, various environmental factors such as surface roughness, vegetation cover, along with the non-robustness of the least squares method. To address this issue, a robust estimation method based on the minimum covariance determinant robust estimation (MCD) is employed for detecting anomalous phases in individual satellites, and we use a low-pass filter (LPF) to correct the outliers. Meanwhile, to address the challenge of poor model generalizability and to comparatively analyze the impact of environmental factors on the prediction results of linear and nonlinear models, we introduce two machine learning methods, namely multiple linear regression (MLR) and extreme learning machine (ELM), to assist the inversion process. The prediction accuracy of each model is thoroughly compared and analyzed.

The subsequent section elucidates the fundamental principle of GNSS-IR soil moisture (SM) estimation grounded in combined GNSS observations. Section 3 provides a description of the genesis, detection, and rectification of abnormal phases. Section 4 discusses the experiments and conclusions in detail, with Section 4.1 offering an exposition of the study area and dataset. Section 4.2 and Section 4.3 delve into the experimental procedure and the formulation of error equations, while Section 4.4 showcases the results of soil moisture (SM) estimation. Section 5 conducts an analysis and discussion of the aforementioned experiments. Finally, Section 6 provides a conclusion to the entire paper.

## 2. GNSS Combined Observations Estimate Soil Moisture Principle

### 2.1. GNSS-IR Near-Surface Forward Reflection Geometry

Ground-based GNSS geodetic receivers not only capture the direct signal transmitted by GNSS satellites but also receive the signal that is reflected from the ground surface. The interference phenomenon occurs between the direct and reflected signals at the center of the receiver antenna, and thus multipath errors are produced and become an essential part of the GNSS observations. The reflected signals received by the antenna primarily come from the first Fresnel reflection zone. Among these, the contribution from signals undergoing specular reflection on the reflecting surface is the most significant. For the sake of convenience in theoretical analysis, only the multipath effect caused by specular reflection is considered. The near-surface forward reflection geometry model is shown in Figure 1.

Based on the geometric relationship depicted in Figure 1, with respect to the direct signal, the reflected signal travels an additional path length, commonly referred to as the path difference. That can be mathematically expressed as a function of variables h and θ,
(1)Δ(t)=2hsinθ(t)
where h refers to the antenna height and θ represents the incidence angle of the direct signal. In addition, compared to the direct signal, the relative time delay δ(t) and the excess phase of the reflected signal $\delta_{\varphi }(t)$, also called the phase delay, are defined as [34],
(2)δ(t)=Δ(t)/c
(3)$$\delta_{\varphi }(t)=2\pi \Delta (t)/\lambda$$
where λ denotes the wavelength of the satellite signal and c is the speed of light. Equation (3) solely considers geometric delay factors, neglecting phase contributions arising from the Fresnel reflection coefficient and antenna radiation pattern. When considering only the specular reflection of signals on the object surface, the direct signal from the satellite, the reflected signal, and the interference signal formed by the superposition of the direct and reflected signal at the antenna can be expressed by the following equations [13],
(4)$$S_{d}(t)=A_{d}\sin(\psi (t))$$
(5)$$S_{m}(t)=A_{m}\sin(\psi (t)+\delta_{\varphi }(t))$$
(6)$$S(t)=S_{d}(t)+S_{m}(t)$$
where $A_{d}$ is the amplitude of the direct signal and $A_{m}$ is the amplitude of the reflected signal, which can also be expressed as $A_{m}=\alpha A_{d}$. In addition, α denotes the amplitude attenuation factor (AAF), which depends on the reflectivity of the reflective surface media and antenna gain pattern [32]. Lastly, the direct phase ψ(t) (unit is radians) is expressed as,
(7)ψ(t)=2π(N+φ(t))
where N and φ(t) are the integer ambiguity and the direct phase, respectively, both in cycles.

### 2.2. Errors Caused by Multipath Effects

When multipath phenomena occur near the GNSS geodetic receiver, β(t) and l(t), the carrier phase multipath error and pseudorange multipath error (unit is meters), can be mathematically expressed as follows [41],
(8)$$\beta (t)=\frac{\lambda }{2\pi }\tan^{-1}\left(\frac{\alpha \sin(2\pi \frac{2h}{\lambda }\sin\theta (t))}{1+\alpha \cos(2\pi \frac{2h}{\lambda }\sin\theta (t))}\right)$$
(9)$$l(t)=\frac{2h\sin\theta (t)\alpha \cos(2\pi \frac{2h}{\lambda }\sin\theta (t))}{1+\alpha \cos(2\pi \frac{2h}{\lambda }\sin\theta (t))}$$
where Δ(t) and $\delta_{\varphi }(t)$ denote the path difference and phase difference between reflected signals and direct signals, respectively, both functions of sinθ(t). In fact, $\alpha =A_{m}/A_{d}\ll 1$ [5], and $1+\alpha \cos(\delta_{\varphi }(t))\approx 1$; in addition, when x≪1, $\tan^{-1}x\approx x$. Equation (9) can be further simplified as:(10)$$\beta (t)=\frac{\lambda }{2\pi }\alpha \sin(\delta_{\varphi }(t))$$
(11)$$l(t)=\alpha \Delta (t)\cos(\delta_{\varphi }(t))$$

Given that the soil moisture and GNSS satellite orbit coordinates are arranged in a continuous time series, the variables (h, l(t), θ(t), τ(t), Δ(t), $\delta_{\varphi }(t)$, α) mentioned earlier are all time-dependent. To streamline the discussion, the time-related terms are omitted from the subsequent text.

### 2.3. The Proposed Combination of Observations

#### 2.3.1. GNSS Observation Equation

In GNSS positioning, when considering only the influence of the atmospheric delays, the observation equations for carrier phase $\varphi_{i}^{s}$ (measured in cycles) and pseudorange $P_{i}^{s}$ (measured in meters) obtained by GNSS geodetic receivers at each epoch can be approximately described as follows [42],
(12)$$\lambda_{i}\varphi_{i}^{s}=d^{s}+c(t_{R}-t^{S})-I_{i}^{s}+T^{s}+\lambda_{i}N_{i}^{s}+\beta_{i}^{s}+\varepsilon_{i}^{s}$$
(13)$$P_{i}^{s}=d_{i}^{s}+c(t_{R}-t^{S})+I_{i}^{s}+T^{s}+l_{i}^{s}+\xi_{i}^{s}$$
where the superscript s is the satellite identifier (PRN), while the subscript i is the frequency band of the GNSS satellite signal. d denotes the geometric distance between the satellite and the receiver, which is generally unknown. $t_{R}$ and $t^{S}$ are the clock deviations of the receiver and the satellite, respectively, and c is the speed of light. $I_{i}^{s}$ represents the ionospheric delay generated during the signal propagation through the ionosphere, which is contingent upon the frequency band. $T^{s}$ denotes the tropospheric delay introduced during the signal propagation through the troposphere, which is unrelated to the signal frequency band and depends on the atmospheric conditions along the path connecting the satellite and the receiver. $\beta_{i}^{s}$ and $l_{i}^{s}$ are the carrier phase and pseudorange multipath errors, respectively. These errors arise due to the influence of multipath effects around the receiving station. $\varepsilon_{i}^{s}$ and $\xi_{i}^{s}$ refer to the random measurement noises carried by the two types of GNSS measurements during the signal propagation, which are independent of the multipath errors.

#### 2.3.2. Linear Combination of Dual-Frequency GNSS Observations

By taking the difference in dual-frequency GNSS observations, the geometric distance $d^{s}$ and the tropospheric delay $T^{s}$ can be cancelled. The geometry-free linear combinations L4 (the carrier phase of two L-frequency bands: L1, L2) and DFPC (the pseudorange of two C/A codes: C1, C2) are frequently available and can be applied to study multipath effects. Solar radiation ionizes a portion of the atmosphere to form the ionosphere, and the electrons generated by this ionization cause a propagation delay known as the ionospheric delay, with its magnitude primarily determined by the frequency. In fact, the first-order component is inversely proportional to the square of the frequency [43]. It is worth noting that ionospheric linear combinations L4 and DFPC are the simple differences in measurement values between two frequencies.
(14)$$\beta_{i,j}^{s}=\lambda_{i}\varphi_{i}^{s}-\lambda_{j}\varphi_{j}^{s}=(-I_{i}^{s}+I_{j}^{s})+(\beta_{i}^{s}-\beta_{j}^{s})+(-\lambda_{i}N_{i}^{s}+\lambda_{j}N_{j}^{s})+(\varepsilon_{i}^{s}-\varepsilon_{j}^{s})$$
(15)$$l_{i,j}^{s}=P_{i}^{s}-P_{j}^{s}=(I_{i}^{s}-I_{j}^{s})+(l_{i}^{s}-l_{j}^{s})+(\xi_{i}^{s}-\xi_{j}^{s})$$
where $\beta_{i,j}^{s}$ and $l_{i,j}^{s}$ denote the combinations of carrier phase and pseudorange measurements at two frequencies (i≠j,j>i), respectively. $-\lambda_{i}N_{i}^{s}+\lambda_{j}N_{j}^{s}$ is the constant term associated with integer ambiguity. Upon eliminating the geometric distance, receiver and satellite clock deviations, and tropospheric delays, the linear combinations L4 and DFPC still include combined ionospheric compositions ($-I_{i}^{s}+I_{j}^{s}$, $I_{i}^{s}-I_{j}^{s}$), multipath error items ($\beta_{i}^{s}-\beta_{j}^{s}$, $l_{i}^{s}-l_{j}^{s}$), and a random measurement noise component ($\varepsilon_{i}^{s}-\varepsilon_{j}^{s}$,$\xi_{i}^{s}-\xi_{j}^{s}$). Compared to the multipath-induced error, the combined ionospheric has a lower sine change, which is considered as the low frequency signal. It can be calculated by a low-pass filter (LPF) and then removed. To extract the multipath, Equation (12) further becomes,
(16)$$\beta_{i,j}^{s}=(\beta_{i}^{s}-\beta_{j}^{s})+(\varepsilon_{i}^{s}-\varepsilon_{j}^{s})$$
(17)$$l_{i,j}^{s}=(l_{i}^{s}-l_{j}^{s})+(\xi_{i}^{s}-\xi_{j}^{s})$$

From Figure 2 and Figure 3, it is evident that there are many small fluctuations in every multipath error series; these will mainly be caused by residual components of ionospheric delay and measurement noise. Additionally, it can be observed that the overall fluctuations of L4 and DFPC multipath errors decrease with an increase in the sine value of satellite elevation angle, indicating that satellite signals at lower elevation angles are more susceptible to multipath effects. Based on this, the satellite elevation angle for GNSS-IR inversion can be deliberately chosen. In our subsequent investigations, multipath errors within the low elevation angle range of 10°−20° were specifically selected for experimentation in order to attain optimal results.

## 3. Generation, Detection and Correction of Abnormal Phases

### 3.1. Generation of Abnormal Phases

The oscillation periods and peak frequencies of combined multipath errors can be determined through Lomb–Scargle spectral analysis [44,45]. Compared to the noise-free scenario, the existence of measurement noise significantly reduces the power spectral density (PSD) at the peak frequency and induces a slight shift in the peak frequency [34]. Generally, the maximum PSD serves as an indicator of the quality of multipath error signals [33]. Meanwhile, the PSD at the main frequency should be at least twice the power of the noise or the second largest frequency in the Lomb–Scargle periodogram (LSP) [24]. Consequently, Lomb–Scargle spectral analysis can be used to evaluate the quality of multipath error signals. For a given reflecting surface, the satellite orbit should have a relatively stable singular frequency [25].

Single frequency observations are influenced by the ionospheric delay, tropospheric delay, receiver and satellite clock deviation, and measurement noise. In the calculation of combined multipath errors for L4 and DFPC, it should be noted that the presence of ionospheric delay is not fully eliminated by the low-pass filter (LPF), and the influence of measurement noise must also be considered. High-quality combined multipath errors may not always be consistently present.

Figure 4 and Figure 5 show the spectral analysis periodograms of L4 and DFPC multipath errors for different satellites. In certain subfigures, the main frequency power of the LSP is twice the secondary frequency power, which indicates a favorable quality of combined multipath errors. However, in the remaining subplots, the main frequency power is not obvious, which may be due to varying influences of slope direction, surface roughness, vegetation canopy around the observation site and random measurement noise on multipath signals from satellite orbits in different orientations. During the experimental period, the occurrence of high-quality multipath errors cannot be guaranteed consistently.

The combined GNSS signals of the noise-affected satellite orbits do not exhibit any prominent frequencies; the low-quality combined multipath errors can be considered observations containing coarse errors. Additionally, the least squares method lacks robustness. If multipath errors containing coarse errors are inputted, the traditional least squares analysis may be destroyed [46]. The calculated results are then the anomalous phase delay. Therefore, effective detection and correction of anomalous phases are of paramount importance in enhancing the accuracy of GNSS-IR estimation of soil moisture.

### 3.2. Detection and Correction of Abnormal Phases

In the GNSS-IR soil moisture estimation process, since the true errors of the observations follow a normal distribution, outliers in the delay phases can usually be identified by integer multiples of the root mean square error. However, this method is vulnerable to the influence of outliers, leading to inaccurate detection results such as missing some outliers or misclassifying normal values as outliers, known as the phenomenon of false negatives and false positives. To address this issue, this study introduces an innovative outlier-detection method based on minimum covariance determinant (MCD) robust estimation. The primary objective of this method is to utilize the Mahalanobis distance and iterative concepts to construct a robust estimate of the covariance matrix. Iterative computation of the robust Mahalanobis distance is performed and tested by applying a Chi-square distribution. This approach facilitates the detection of anomalies within the dataset and assigns variable weights to each detected anomaly.

For a dataset consisting of n samples with a known dimensionality, a random sample with a size of m is selected from the dataset. Generally, the ability of the MCD robust estimation method to handle outliers increases as the value of m decreases. When there is an excessively small value for m, it difficult to distinguish abnormal values from normal ones. So, the default value of m is typically set to 0.75n. At this point, the mean and covariance matrix of the sample is the initial sample mean and covariance matrix; the formula can be expressed as,
(18)$$u_{1}=\frac{1}{h}\sum_{i=1}^{h}X_{i}$$
(19)$$S_{1}=\frac{1}{h}\sum_{i=1}^{h}(X_{i}-u_{i}){\left(X_{i}-u_{i}\right)}^{T}$$

Equations (18) and (19) are used to solve the Mahalanobis distance between the dataset and the mean of the selected sample, which is expressed as follows,
(20)$$MD(i)=\sqrt{{\left(X_{i}-u_{i}\right)}^{T}S_{1}^{-1}(X_{i}-u_{i})}$$

From Equation (20), n Mahalanobis distances are computed and sorted in descending order, and h sample data with minimum distances are selected. Based on the sample, the mean estimate and covariance matrix estimate are recalculated. Then, we repeat the above process, and the iteration is stopped when $\det(S_{i-1})=\det(S_{i})$ or $\det(S_{i})=0$. The covariance matrix determinant in the iterative process follows the following relationship,
(21)$$\det(S_{1})\geq \det(S_{2})\geq \ldots \geq \det(S_{i-1})\geq \det(S_{i})$$

According to Equation (21), the iterative process described above is proven to converge. The mean and covariance matrix obtained at the termination of the iteration represent the robust mean and robust covariance, respectively. By substituting these values into the Mahalanobis distance equation, the Mahalanobis distances of the dataset can be obtained, which follows a Chi-square distribution with p degree of freedom. If Equation (22) is satisfied, the sample point can be identified as an outlier; otherwise, it is considered a normal value,
(22)$$MD(i)>\sqrt{\chi_{p,\alpha }^{2}}$$

The utilization of MCD has been employed to detect the locations of outliers and remove them. To maintain the temporal continuity of the delay phases, while preserving its high quality, a moving average filter (MAF) is applied. This approach is executed to correct anomalous phases, employing MATLAB software’s built-in smooth function. Performing these operations ensures that more satellites possess high-quality delay phases, which provides a broader selection of satellites for subsequent experiments.

## 4. Experiments and Results

### 4.1. Experimental Datasets

The experimental data for this paper were obtained from the Plate Boundary Observatory (PBO) GNSS station: the P031 station, which is part of the Earth Scope project in the United States. GNSS data can be accessed online at any time (https://www.unavco.org/data/dai/) (accessed on 25 March 2023). The P031 station is situated in Colorado, USA, at coordinates 39.5155° N, 107.9087° W. The data span from 8 March to 30 May in 2017, consisting of carrier phase and pseudorange measurements in C/A code for L1, L2, and L5 frequency bands. For the purpose of this study, only the L1 and L2 carrier phase and pseudorange measurements are utilized to retrieve the variations in the soil moisture. The P031 station is located in an open area without any building obstruction, and the surrounding terrain is relatively flat with well-preserved land integrity. The on-site environmental conditions around the station and the status of the receiver antenna during the data collection period are shown in Figure 6.

The receiver antennas of the majority of PBO GNSS stations are positioned at a vertical distance of approximately 1.8 m from the ground [23]. The focus of the research is to analyze the variation trends of amplitude decay factor α and delay phase $\delta_{\varphi }$. When determining the initial values of the characteristic parameters, the receiver antenna height default is 1.80 m to investigate the effect of soil moisture variation on the delay phase. The summary of GNSS receiver parameters for station P031 is presented in Table 1.

Since obtaining the true soil moisture value near the station is unfeasible, the reference value of soil moisture is estimated from the SNR of the GPS L2C frequency band signal. The soil moisture at PBO stations is derived from the daily average of station-level soil moisture, and the median of soil moisture values estimated from all satellite orbits for each day is used [24]. The soil moisture values at PBO stations can be downloaded at (https://cires1.colorado.edu/portal/) (accessed on 26 March 2023).

Most reflected signals pass through the first Fresnel zone, which is a set of ellipses determined by the satellite azimuth angle, satellite elevation angle, and receiver antenna height. The center of the first Fresnel zone is at,
(23)$$S_{x}=h\cot\theta$$
(24)$$S_{y}=0$$
where $S_{x}$ represents the projection position of the center of the ellipse on the surface, and $S_{y}$ denotes the ground projection position of the receiver.

Since the signals captured by the antenna within the first Fresnel zone have the maximum energy, the GNSS-R technology primarily focuses on the first Fresnel zone. The mathematical expressions for the semi-major R (aligned along the direction to the GPS satellite orbit) and semi-minor r axes of the improved ellipses in this zone are [47],
(25)$$R=\frac{\sqrt{\lambda h\sin\theta }}{\sin^{2}\theta }$$
(26)$$r=\frac{\sqrt{\lambda h\sin\theta }}{\sin\theta }$$
where λ is the wavelength of the signal band; θ is the satellite elevation angle; and h is the vertical distance from the antenna phase center to the reflecting surface.

As shown on the right side of Figure 4, the area surrounding the P031 station is flat plain terrain, with no other signal sources within a 30 m radius around the station and sparse surface vegetation. Interference from external signal sources and the effects of vegetation coverage on the experiment are excluded. Figure 7 shows the first Fresnel reflection zone at different satellite elevation angles. According to calculations, when the satellite elevation angle is 10°, the distances from the reflection points in various directions to the measurement station do not exceed 19 m in the first Fresnel reflection zone. When the satellite elevation angle is 20°, the first Fresnel reflection zone moves closer to the measurement station, and the distances from the reflection points in various directions to the station do not exceed 7 m. Therefore, the reflected signals within the first Fresnel reflection zone of 10°−–20° can be considered as ideal data sources for retrieving soil. This paper does not put too much emphasis on the satellite azimuth.

### 4.2. Experimental Technical Route

The technical flowchart of the soil moisture estimation used in this study is shown in Figure 8, which can be roughly divided into four parts.

(1)Data preprocessing: Preprocessing of the observation (OBS) files and navigation (NAV) files obtained from the GNSS receiver. The TEQC software was used to extract essential parameters such as carrier phase observation, pseudorange observation, satellite elevation angle, satellite azimuth angle, and epoch information.(2)Estimation of characteristic parameters: We constructed dual-frequency carrier phase and pseudorange linear combinations L4 and DFPC, respectively. The low-pass filter (LPF) was applied to remove the combined ionospheric delay and then we calculated combined multipath errors. By determining the initial values of path difference, delay phase, and amplitude attenuation factor, we established the indirect leveling error equation. Meanwhile, we extracted five combined multipath errors from the optimal elevation angle range of different effective satellites for participation in the calculation of single-satellite characteristic parameters.(3)Detection and correction of abnormal phases: The MCD method was employed to identify the locations of abnormal values in the delay phase. Subsequently, the detected abnormal phases were corrected and replaced using the moving average filter (MAF). Detection, correction, and replacement of abnormal delay phases of each satellite was conducted using MCD and MAF in succession. To reflect the effectiveness of the above methods for outlier detection and correction, Pearson correlation coefficients (R) were calculated separately for the delay phase before and after correction in relation to soil moisture.(4)Soil moisture estimation: The corrected multi-satellite delay phases and the corresponding soil moisture were divided into training and prediction sets. The training set was employed for model training to establish linear and nonlinear models, while the prediction set was used to estimate soil moisture and assess the accuracy of the model predictions.

**Figure 8 sensors-23-07944-f008:**
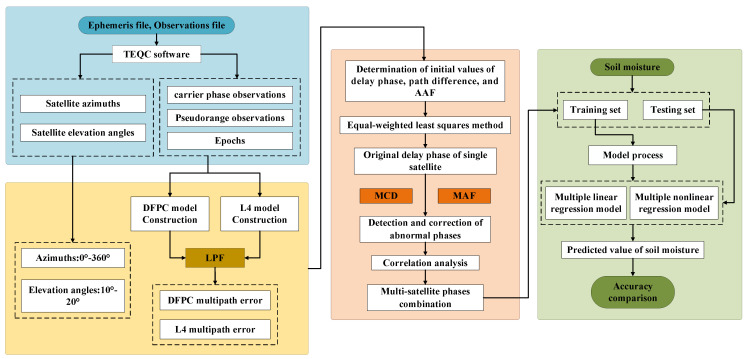
Technical flowchart for GNSS dual-frequency combined measurements for soil moisture estimation. Note that TEQC (Translation Editing and Quality Checking) is a GNSS data preprocessing software developed by the UNAVCO Facility as a public freeware service for GPS monitoring to support station data management in geological research (https://www.unavco.org/software/data-processing/teqc/) (accessed on 26 March 2023).

### 4.3. Error Equation Establishment and Parameter Solving

Based on Equations (1) and (3), it can be deduced that the wave delay Δ and the delay phase $\delta_{\varphi }$ are both functions of the antenna height h. Over a certain period, different values of soil moisture will result in varying antenna heights of the receiver, which, in turn, will cause changes in the wave delay and delay phase. However, the trend consistency between soil moisture and the delay phase is higher than that between soil moisture and the antenna height [25]. Furthermore, from the perspective of microwave reflection mechanisms, the variability of soil moisture can also lead to changes in soil dielectric constant and even surface reflectivity, thereby affecting the variability of the amplitude fading factor. Therefore, in order to calculate the path difference, phase delay, and AAF, and validate their ability to reflect the soil moisture variability, the nonlinear observation equation is linearized to obtain the error equation,
(27)$$l+V_{l}=\alpha_{0}\Delta_{0}\cos\delta \varphi_{0}+\Delta_{0}\cos\delta \varphi_{0}V_{\alpha }+\alpha_{0}\cos\delta \varphi_{0}V_{\Delta }-\alpha_{0}\Delta_{0}\sin\delta \varphi_{0}V_{\delta \varphi }$$
(28)$$\beta +V_{\beta }=\alpha_{0}\sin\delta \varphi_{0}+\sin\delta \varphi_{0}V_{\alpha }+\alpha_{0}\cos\delta \varphi_{0}V_{\delta \varphi }$$
where observations l and β are readily calculated through Equations (14) and (15), and $V_{l}$ and $V_{\beta }$ denote the infinitesimal adjustments of the L4 and DFPC multipath errors, also known as the corrections to the observations l and β. $\alpha_{0}$ represents the initial value of the amplitude attenuation factor, and the amplitude $A_{m}/A_{d}$ is required to calculate l and β. The direct signal amplitude usually defaults to 1, yet the reflected one is equal to the product of the antenna gain ratio between the positive and negative elevation angle and the media surface reflectivity (0.3–0.8) [13]. Since the primary focus of this paper is to explore the relationship between delay phases and soil moisture, the value of $\alpha_{0}$ is set to 0.3 when ignoring the influence of antenna gain. The initial value of the path difference, $\Delta_{0}$, can be deduced by the antenna height h and satellite elevation angle θ using Equation (1). $\delta_{\varphi_{0}}$ represents the initial value of the delay phase, which can be calculated by the wavelength λ and path difference $\Delta_{0}$ previously mentioned in Equation (3). $V_{\alpha }$, $V_{\Delta }$, and $V_{\delta \varphi }$ are the correction terms for the corresponding estimated parameters, namely α, Δ, and $\delta_{\varphi }$ (L4: $V_{\alpha }$, $V_{\delta \varphi }$; DFPC: $V_{\alpha }$, $V_{\Delta }$, $V_{\delta \varphi }$).

According to Equations (8), (9), (16) and (17), it is known that L4 and DFPC multipath errors are functions of satellite elevation angle θ. In this study, the least squares method is employed for estimating the characteristic parameters. Owing to significant multipath error fluctuations within a wide range of altitude angles, opting for a longer combined multipath error sequence for parameter estimation may not yield the most accurate parameter values. The surplus multipath errors shall be regarded as observations encompassing coarse errors, concurrently ensuring that the length of the multipath error series exceeds the count of parameters to be estimated, thus securing a singular solution for the equation. Consequently, in this study, we selected a dual-frequency combined multipath error sequence with a length of 5 for parameter estimation. We assumed that the path difference and delay phase remain constant over a short time period while ignoring the minor fluctuations of soil moisture.

Thus, we selected a length of 5 for the multipath errors, along with the corresponding satellite elevation angle sequence. Assigning equal weights to each data point, indirect leveling error equations were established to solve the optimal estimated values of the parameters. It should be noted that when determining the initial values of the path difference and delay phase, the calculation results based on the first value of the satellite elevation angle sequence should be followed.

Currently, all GPS satellites are capable of transmitting dual-frequency signals. However, when considering the quality of L4 and DFPC multipath errors, as well as the ability of the receiver to simultaneously record dual-frequency pseudorange and carrier phase observations during the experimental period, the number of available satellites may be limited. Figure 9 displays the elevation angle variation trend of all GPS satellites transmitting dual-frequency pseudorange and carrier phase observations that could be received at station P031 on 10 April 2017 (DOY: 100) throughout the day. The observable time periods differ significantly due to variations in the orbital azimuth of each satellite, typically ranging from 1.0 to 2.5 h. Satellites at elevation angles between 10° and 20° have shorter observable time periods, generally ranging from 0.5 to 1.0 h.

In previous studies, satellite signals in the range of 5°–30° were usually selected for experiments. Nonetheless, owing to the stochastic noise components inherent in signal transmission, the combined multipath errors of the same satellite within varying altitude angle ranges exhibit distinct quality profiles in the power spectral density periodogram. To ensure the involvement of more available satellites in the experimental process while concurrently obtaining high-quality combined multipath error data, it becomes imperative to judiciously define the altitude angle range for satellite selection. Combined multipath errors with low satellite altitude angles have more noise components, and the combined multipath errors with high satellite altitude angles with non-significant multipath effects lack surface parameter information. Consequently, following an extensive series of experiments, this paper narrows down the satellite altitude angle range to a constrained 10° to 20°. In Figure 7, a few GPS satellites remain at lower elevation angles for a certain observation time period. For example, PRN9 remains below 10° from 04:30 to 06:12, and PRN12 remains below 10° from 12:50 to 15:10. Throughout the experimentation, the multipath errors within this elevation range will be excluded.

Based on the analysis above, we set the satellite cutoff elevation angle to 10°. Due to variations in the quality of L4 and DFPC multipath errors on different satellite orbits, the optimal elevation angle range may vary among different satellites. Taking 10 April 2017 as an example, the optimal satellite elevation angle ranges for estimating characteristic parameters of different satellites were determined through extensive experiments, as shown in Table 2.

### 4.4. Soil Moisture Estimates

Without removing satellites with poor multipath quality, all available satellites that could be continuously observed during the experiment period were selected (PRN1, PRN3, PRN6, PRN8, PRN10, PRN25, PRN26, PRN27, PRN30, PRN32). The DFPC and L4 multipath errors within a satellite elevation sequence of length 5 were used as observations, and the best estimates of delay phase, path difference, and AAF were calculated through the principle of equal weight least squares. This paper focuses on the response of delay phases to changes in soil moisture, with a temporal resolution in days. Consequently, the influences of satellite motion, soil moisture and other environmental factors within a short period of time are ignored. For a detailed description of the process for calculating characteristic parameters using the L4 and DFPC combined signals, please refer to Appendix A.

For a single satellite, when the quality of L4 and DFPC multipath errors is subpar, the corresponding delay phase is regarded as an outlier. The presence of a certain number of outlier phases hinders the accurate retrieval of soil moisture. To ensure the continuity of the single-satellite delay phase, outliers are not directly eliminated, but rather detected and corrected through appropriate procedures. The effective satellite initial delay phases were used as the sample dataset for the MCD method, which was divided into training and test sets according to a certain ratio. The confidence level α was set to 0.975 and the degree of freedom p took the value of 1, resulting in $\sqrt{\chi_{1,0.975}^{2}}$ = 2.2414. When the Mahalanobis distance MD(i) of a sample point in the dataset was greater than 2.2414, it was considered as an abnormal phase value. In such cases, we applied the MAF to correct outliers. On the daily time scale, the initial delay phase and the corrected delay phase for both L4 and DFPC methods were obtained. To assess the relationship between soil moisture and delay phases at station P031, Pearson correlation coefficients were calculated and plotted as histograms, as shown in Figure 8.

As can be discerned from Figure 10a,b, the combination of MCD and MAF can enhance the quality of the delay phase. Upon detection and correction of outliers, the R of both DFPC and L4 methods saw varying degrees of improvement. For the DFPC method, approximately 63% of satellites achieved an R greater than 0.7, with a maximum value of 0.74. In comparison, the L4 method had 81% of satellites with an R greater than 0.7, reaching a maximum value of 0.77. The average R across 11 satellites for the DFPC and L4 methods were 0.66 and 0.70, respectively. Furthermore, it is apparent that combined multipath errors for different satellites differ in their ability to perceive surface soil moisture. This difference is due to how signals from different satellites are reflected by the Earth’s surface and contain soil moisture information of the surface in different orientations. Relying solely on the characteristic parameters of a single satellite orbit would not accurately and effectively monitor soil moisture within the effective range of the first Fresnel reflection zone. To further improve the performance of soil moisture estimation using L4 and DFPC multipath errors, two models, namely multiple linear regression (MLR) and extreme learning machine (ELM), were used for multi-star fusion experiments. The in situ soil moisture values and each effective satellite delay phase at site P031 over 84 consecutive days were used as samples, with a training-to-testing data split ratio of approximately 6:1. The first 72 days of data were employed for model establishment and the subsequent 12 days of data were employed for prediction. The comparison chart of retrieval errors of soil moisture in different cases is shown in Figure 11, while Figure 12 and Figure 13 provide the comparative analysis of soil moisture prediction results from both the combination method and model perspective, respectively.

For the L4 and DFPC combination methods, the Pearson correlation coefficients between the predicted soil moisture values of the MLR model and the true soil moisture values of measurement station P031 were 0.81 and 0.84, respectively, while those between the predicted soil moisture values from the ELM model and the true soil moisture values at site P031 were as high as 0.88 and 0.90. To comprehensively reflect the performance of the above two models for estimating soil moisture, we further evaluate the accuracy indexes of each model, as presented in Table 3.

## 5. Discussions

When estimating in situ soil moisture at the site using DFPC and L4 multipath errors, under the assumption of not considering the influence of surface environmental factors on soil moisture, the MLR model can accurately represent the linear regression relationship between the multi-satellite delay phase and soil moisture. Due to slight differences in topography, surface roughness, and vegetation coverage thickness in different directions around the site, the combined effect of multiple factors causes the relationship between soil moisture and the multi-satellite corrected delay phase to be inaccurately represented by a linear regression model; therefore, we tried using the ELM model to represent the nonlinear regression relationship between them.

As can be seen from Figure 9, for the same combination of observations, the sample estimation errors of the ELM model were smaller than those of the MLR model. In the case of different combinations of observations, the performance of the ELM model remained relatively stable, with retrieval errors consistently smaller than the MLR model. Figure 10 and Figure 11 provide a direct illustration of the prediction accuracy under different combination methods and models. Both the DFPC and L4 methods yielded high-accuracy soil moisture retrieval, and the ELM model demonstrated a better fit between predicted values and actual soil moisture values compared to the MLR model. As can be seen in Table 3, the predicted soil moisture values obtained using the DFPC-MLR and DFPC-ELM methods exhibited Pearson correlation coefficients of 0.81 and 0.88, respectively, with the true in situ soil moisture values at the site. However, the RMSE, MAE, and STD predicted using the DFPC-ELM method were smaller than those of the DFPC-MLR method. The predicted soil moisture values obtained using the L4-MLR and L4-ELM methods were also correlated with the true in situ soil moisture values at the site, with Pearson correlation coefficients of 0.84 and 0.90, respectively, which were slightly higher than those of the DFPC-MLR and DFPC- ELM methods. Additionally, the RMSE, MAE, and STD predicted by the L4-MLR model were smaller than L4-ELM model. In summary, the linear regression model MLR cannot compensate for the effects of the above-mentioned multiple factors on soil moisture variation, while the non-linear regression model ELM exhibits better predictive capability for soil moisture at the site. Simultaneously, the single-satellite orbit characteristic parameters are unable to accurately estimate soil moisture information around the measurement station. By combining orbit characteristic parameters from different satellite orientations, the soil moisture information of the site can be estimated from different directions. This approach not only reduces the impact of surface environmental factors on soil moisture estimation but also mitigates the influence of residual stubborn anomalous delay phases on the experiment to some extent.

To better characterize the capability of L4 and DFPC multipath errors to estimate soil moisture, we employed the error propagation law to calculate and compare their respective standard deviations. The measurement error of GPS C/A code three-frequency carrier phase observations has a same standard deviation of $\sigma_{L}=1\;\mathrm{mm}$, and the accuracy of GPS C/A code three-frequency pseudorange observations is relatively low, at $\sigma_{P}=2.93\;m$. Therefore, the standard deviations of L4 and DFPC multipath errors were $\sigma_{L4}=\sqrt{2}\sigma_{L}=1.41\;m$ and $\sigma_{P4}=\sqrt{2}\sigma_{P}=4.14\;m$, respectively. Compared to the L4 method, DFPC multipath errors are of poorer quality, which is one of the reasons for the presence of more abnormal phases in the single-satellite delay phase sequence of the DFPC method. By employing the minimum covariance determinant robust estimation method to detect and correct abnormal phases, higher-quality and continuous delay phases can be obtained. This, in turn, leads to desirable values of mean absolute error (MAE), root mean square error (RMSE), and standard deviation (STD) when predicting soil moisture.

When determining the initial values of the characteristic parameters, the default amplitude attenuation factor is 0.3. The delay phase and path difference are calculated based on the first value of the satellite elevation angle sequence. Alternatively, first, the delay phase sequence and path difference sequence must be calculated separately, and then the mean values of the respective sequences are defined as the initial values for the delay phase and path difference.

## 6. Concluding Remarks

Based on the analysis of the near-surface forward reflection principle of GNSS-IR and the combination method of dual-frequency GNSS observations, this paper proposes two new approaches for estimating soil moisture: the dual-frequency pseudorange combination (DFPC) method and the dual-frequency carrier phase combination (L4) method. The main advantages of these two methods are the construction of combined multipath error models instead of SNR for soil moisture estimation, as well as the consideration of detection and correction of abnormal phases. The DFPC and L4 methods were utilized to construct linear combinations of dual-frequency observations from GPS, which excluded geometric range and tropospheric delay. Subsequently, a low-pass filter (LPF) was employed to attenuate the combined ionospheric errors. Soil moisture estimation was then achieved through indirect adjustment within a narrow elevation angle range. Notably, this study marks the first application of the DFPC method for estimating soil moisture information. The methods proposed in this study are thoroughly validated using the true soil moisture values obtained from the station P031 of the Plate Boundary Observatory (PBO) H_2_O project. Since both the DFPC and L4 methods were geometry-free and non-tropospheric combinations, and we assumed minimal variation in combined ionospheric delay within short time epochs, both methods achieved high Pearson correlation coefficients (R) and accurate soil moisture estimation precision.

(1)Based on the carrier phase and pseudorange observation equations, it is known that after eliminating the effects of tropospheric delay, geometric distance factors, and ionospheric delay from the combination of GNSS dual-frequency observation values, the random noise components in the combined observations can result in anomalous feature parameters. To enhance the quality of the feature parameters and maintain the continuity of the feature parameter sequence, abnormal delay phases of all valid satellites need to be detected using the MCD method before modeling. Additionally, a moving average filter (MAF) should be used to correct the detected anomalies, resulting in a high-quality and continuous delay phase sequence. Subsequent to this correction, the correlation between the delay phase of each available satellite and soil moisture is improved to varying degrees compared to before the correction.(2)Both DFPC and L4 multipath errors can serve as substitutes for SNR in soil moisture retrieval, thereby enriching the data sources available for GNSS-IR. In the process of fusing data from multiple satellites to estimate soil moisture, the MLR and ELM models integrate multi-satellite phase delays from linear and nonlinear perspectives, respectively, and both models yield commendable prediction accuracies. However, it is worth noting that the accuracy of the ELM model surpasses that of the MLR model. This phenomenon can be attributed to the slight variations in environmental factors surrounding the measurement station in different directions. These variations result in a nonlinear functional relationship between soil moisture and multi-satellite phase delays, which is more effectively captured by the ELM model. In contrast, the linear function employed by the MLR model is less adept at representing this intricate relationship.(3)Compared to traditional SNR methods, when estimating the delay phase using DFPC and L4 multipath error sequences, there is no need for input signals within a large elevation angle range or analyzing the main frequency of multipath error signals. The computation of the combined multipath error requires only the selection of successive calendar elements with both elevation angle and multipath error information required for the calculation. With access to high-sampling-rate ground truth data on soil moisture for validation, this approach can achieve soil moisture estimation at high temporal resolution under the GPS system, with time resolution significantly improved to nearly an hour. As a result, it enables accurate and dynamic prediction of soil moisture with exceptional precision.

At present, GNSS satellites possess the capability to transmit dual-frequency signals. It should, however, be noted that L4 and DFPC are inter-frequency differenced combinations of observations, which may amplify the random measurement noise of a single-frequency signal. In future experiments, it is worth considering the results obtained from combining single-frequency carrier phase and pseudorange measurements for comparative purposes. Not every orbit of the available satellites can accurately estimate soil moisture information, and how to select the best satellite orbit combination needs to be further explored. In constructing the geometry-free and ionosphere-free combinations of the measurements of GNSS dual-frequency signals, the combined multipath errors are still influenced by the combined ionospheric delay. To address this, a low-pass filter (LPF) was employed in this study to eliminate the ionospheric trend term. In future research, alternative methods such as a high-order polynomial (HOP) or Savitzky–Golay filter could be explored and compared in terms of their performance. Furthermore, as this experiment focused on multipath errors within a relatively narrow range of elevation angles for estimating characteristic parameters, future investigations could integrate multi-system and multi-satellite dual-frequency signals to enable dynamic monitoring of soil moisture with high temporal resolution. Drawing upon the collection of environmental variable data from various directions surrounding the station, forthcoming analyses should emphasize the impact of individual environmental variables on soil moisture estimations and their interrelationships. This advancement would further enhance the practical value of GNSS-IR microwave remote sensing technology.

## Figures and Tables

**Figure 1 sensors-23-07944-f001:**
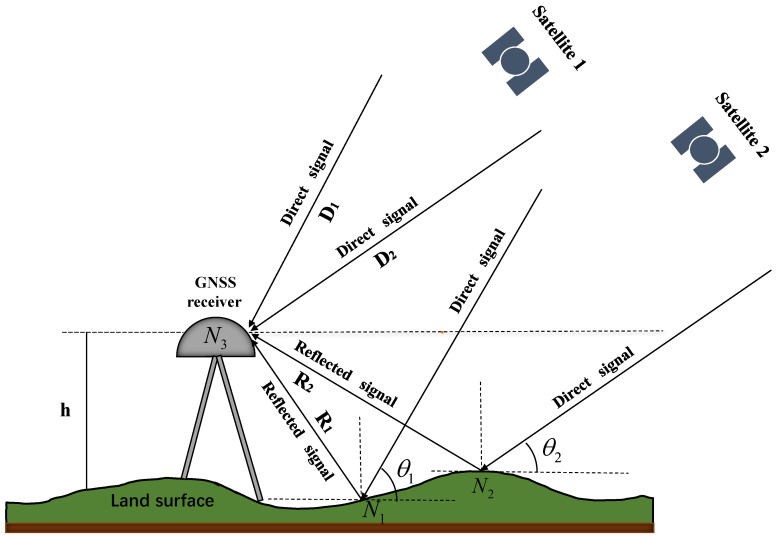
The near-surface forward reflection geometry model for GNSS-IR, where the right-hand circularly polarized (RHCP) antenna receives the direct signal from the satellite, as well as the reflected signal from the ground after the direct signal undergoes reflection. $N_{1}$ and $N_{2}$ represent the positions of satellite signals undergoing specular reflection from satellite 1 and satellite 2, respectively. $N_{3}$ denotes the GNSS measurement-type receiver antenna, h is the vertical distance from the antenna center to the reflecting surface. $\theta_{1}$ and $\theta_{2}$ are the elevation angles of satellite 1 and satellite 2, respectively, as seen from the receiver antenna. $D_{1}$ and $D_{2}$ represent the direct signals received by the receiver antenna from satellite 1 and satellite 2, respectively. $R_{1}$ and $R_{2}$ denote the reflected signals from satellite 1 and satellite 2, respectively, after undergoing reflection on the ground.

**Figure 2 sensors-23-07944-f002:**
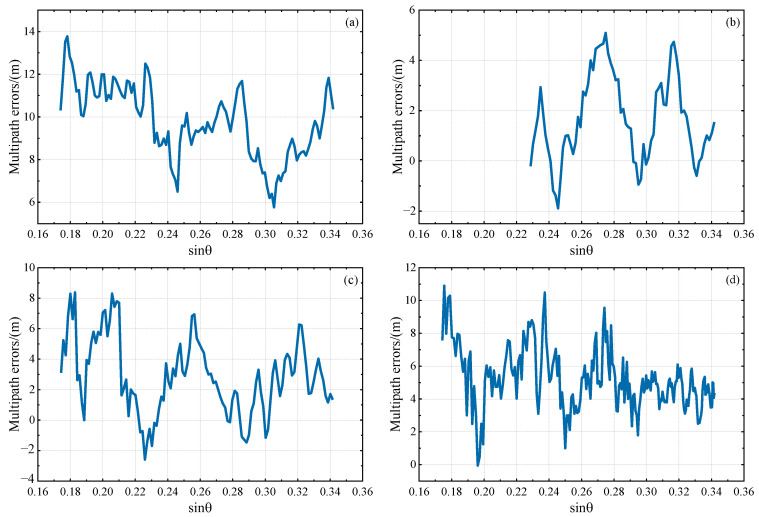
The variation trend of DFPC multipath error with the sine value of satellite elevation angle in the low elevation angle range of 10°−20°. (**a**) GPS PRN1 descending segment, (**b**) GPS PRN3 ascending segment, (**c**) GPS PRN6 descending segment, (**d**) GPS PRN25 descending segment.

**Figure 3 sensors-23-07944-f003:**
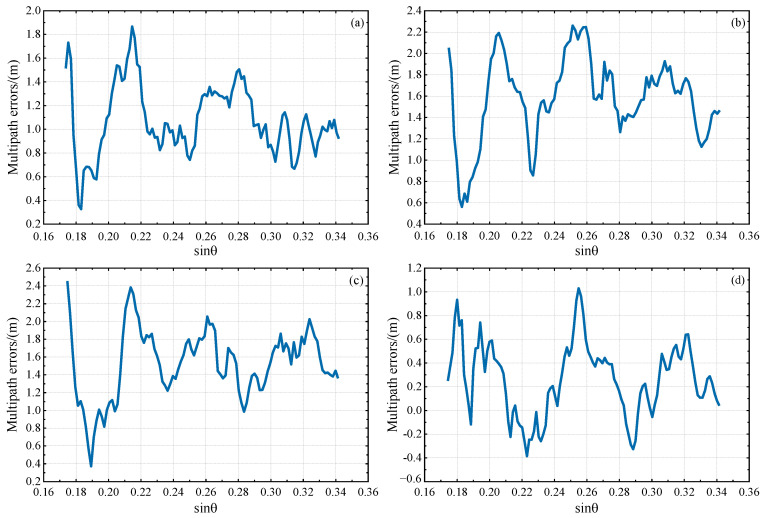
The variation trend of L4 multipath error with the sine value of satellite elevation angle in the low elevation angle range of 10°−20°. (**a**) GPS PRN27 descending segment, (**b**) GPS PRN26 descending segment, (**c**) GPS PRN8 descending segment, (**d**) GPS PRN6 descending segment.

**Figure 4 sensors-23-07944-f004:**
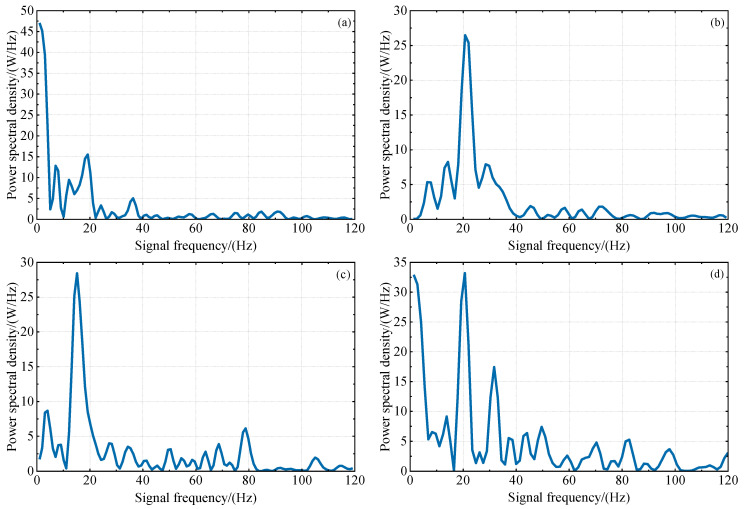
Lomb–Scargle spectral analysis of DFPC multipath errors within the satellite elevation angle range of 10°−20°, using P031 station as an example. The four subplots represent the spectral analysis periodograms for different satellites: (**a**–**d**) PRN1, PRN3, PRN6, PRN25. The horizontal and vertical axes represent the frequency (F) and power spectral density (PSD) of the multipath errors, respectively.

**Figure 5 sensors-23-07944-f005:**
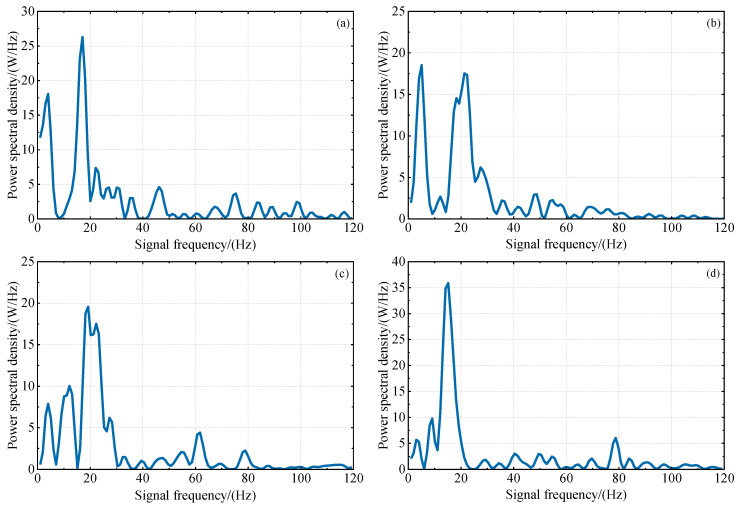
Lomb–Scargle spectral analysis of L4 multipath errors within the satellite elevation angle range of 10°−20°, using P031 station as an example. The four subplots represent the spectral analysis periodograms for different satellites: (**a**–**d**) PRN27, PRN26, PRN8, PRN6. The horizontal and vertical axes represent the frequency (F) and power spectral density (PSD) of the multipath errors, respectively.

**Figure 6 sensors-23-07944-f006:**
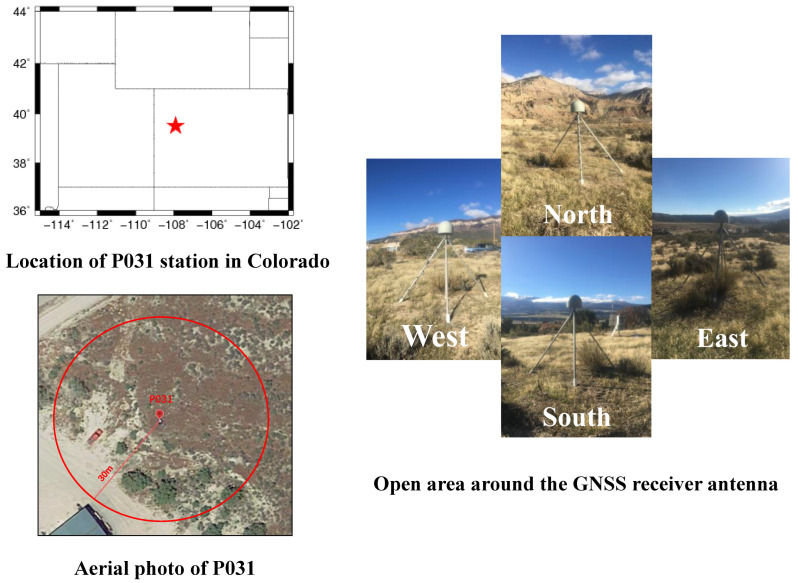
The geographic location of the P031 observation station is in the top left corner of the image, the station is indicated by a red pentagram. The bottom left corner shows an aerial orthoimage, indicating that the station is in an open area. The right side of the image shows the site surroundings and receiver condition from four directions: southeast, southwest, northwest, and northeast. The station is equipped with a SEPT POLARX5 receiver and a TRM59800.00 antenna.

**Figure 7 sensors-23-07944-f007:**
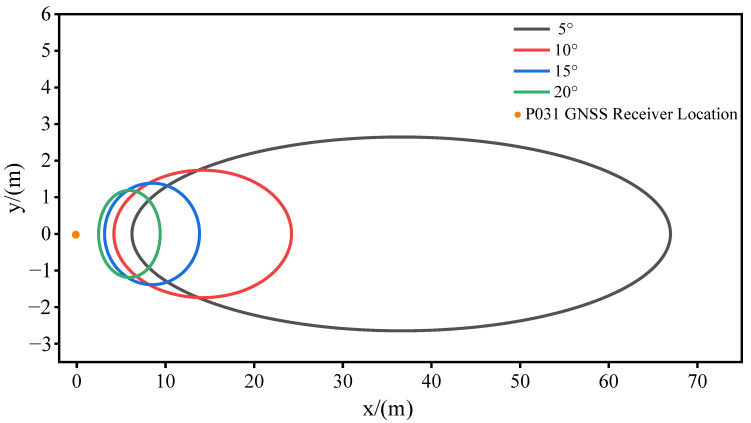
The trajectory of the first Fresnel reflection zone at satellite elevation angles of 5°, 10°, 15°, and 20° (take L2 frequency band signals as an example).

**Figure 9 sensors-23-07944-f009:**
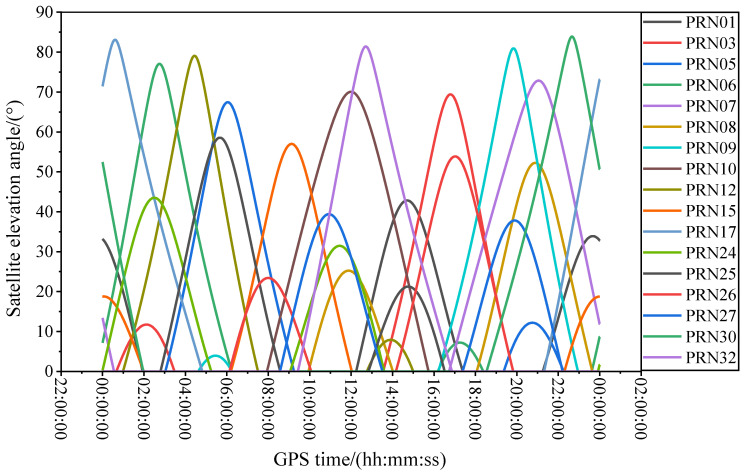
Variation trend of elevation angles for almost all GPS satellites that can be observed simultaneously with dual-frequency pseudorange (C1, C2) and dual-frequency carrier phase (L1, L2) measurements within a single day (P031, 10 April 2017, DOY: 100).

**Figure 10 sensors-23-07944-f010:**
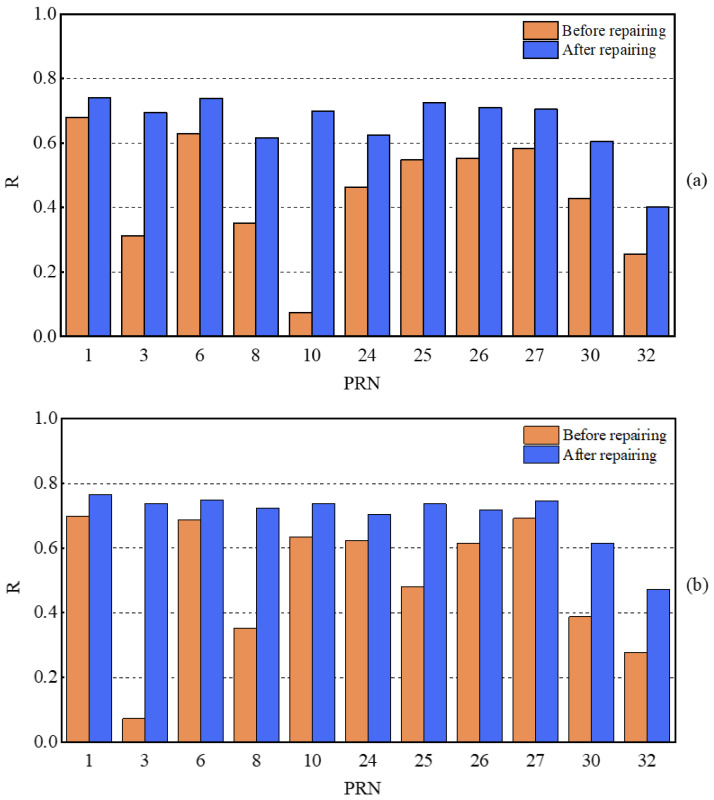
Histograms of Pearson correlation coefficients between the delay phases of all available GPS satellites and observed soil moisture values. (**a**) DFPC method before and after correction of outliers, and (**b**) L4 method before and after correction of outliers.

**Figure 11 sensors-23-07944-f011:**
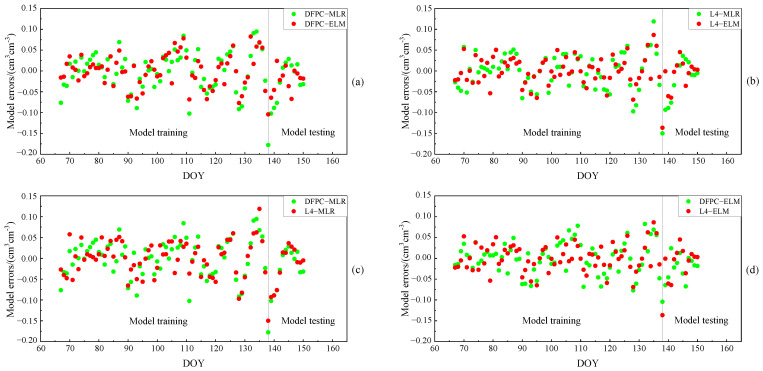
The retrieval error distribution of soil moisture. (**a**) The DFPC method for two models, (**b**) the L4 method for two models, (**c**) the MLR model in different methods, (**d**) the ELM model in different methods. Note that the temporal resolution of the estimation is one value per day.

**Figure 12 sensors-23-07944-f012:**
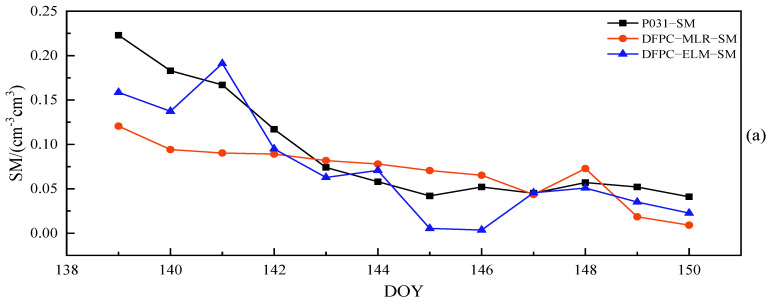
For each combination method, comparison line charts of predicted soil moisture values of the MLR and ELM models against measured (ground-truth) soil moisture values at station P031. (**a**) The DFPC method and (**b**) the L4 method.

**Figure 13 sensors-23-07944-f013:**
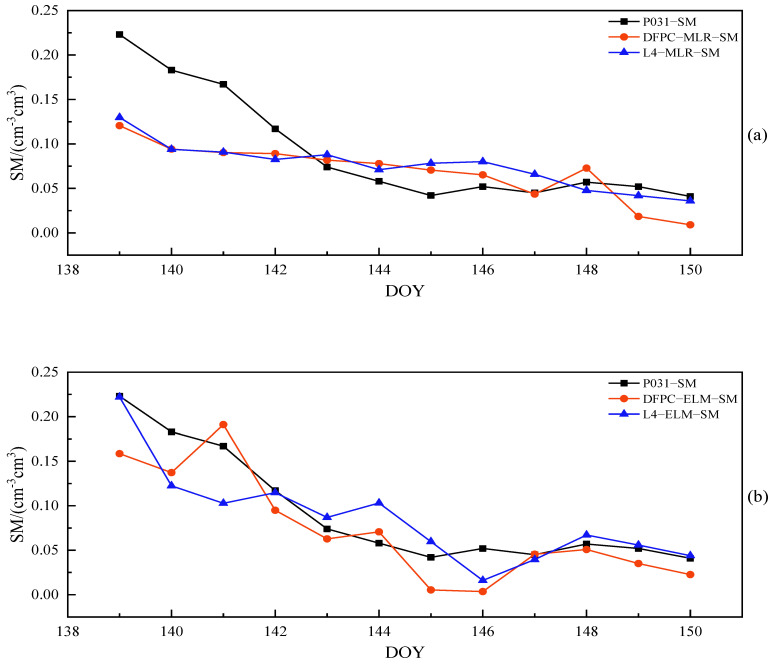
For each model, comparison line charts of predicted soil moisture values of the DFPC and L4 methods against measured (ground-truth) soil moisture values at station P031. (**a**) The MLR model and (**b**) the ELM model.

**Table 1 sensors-23-07944-t001:** The antenna parameters for the GNSS geodetic receiver at station P031.

Item	Parameters
GNSS receiver type	SEPT POLARX5
Sampling interval	15.0 s
Antenna gain pattern	TRM59800.00
Antenna center height	1.8 m

**Table 2 sensors-23-07944-t002:** Selection results of elevation angles, azimuth angles, and corresponding observation times for each available GPS satellite on 10 April 2017 (DOY: 100) at station P031.

GPS Satellite Number (PRN)	GPS Time Range/(hh:mm:ss)	Elevation Angle Range/(°)	Azimuth Angle Range/(°)
PRN1	14:24:00–14:25:00	13.578–13.96	319.221–319.145
PRN3	15:55:45–15:56:45	18.041–18.416	307.636–307.789
PRN6	7:3:45–7:4:45	14.796–14.452	59.2–59.4
PRN8	14:52:45–14:53:45	11.48–11.22	254.51–254.147
PRN10	16:49:45–16:50:45	10.757–10.377	133.761–133.97
PRN24	14:38:00–14:39:00	11.028–10.687	39.3–39.1
PRN25	9:22:30–9:23:30	16.796–16.408	205.4–205.2
PRN26	10:49:00–10:50:00	12.763–12.52	259.341–258.964
PRN27	14:11:45–14:12:45	16.694–16.368	232.956–232.716
PRN30	20:54:45–20:55:45	14.674–14.978	273.4–273.7
PRN32	17:16:30–17:17:30	20.891–20.58	82.549–82.885

**Table 3 sensors-23-07944-t003:** The statistical results of the accuracy indices for soil moisture prediction of MLR and ELM models using the DFPC and L4 methods, respectively.

Method	Model	R	RMSE/(cm^−3^cm^3^)	STD/(cm^−3^cm^3^)	MAE/(cm^−3^cm^3^)
DFPC	MLR	0.81	0.051	0.049	0.038
	ELM	0.88	0.036	0.034	0.027
L4	MLR	0.84	0.049	0.047	0.036
	ELM	0.90	0.033	0.031	0.021

The accuracy indices include RMSE (root mean square error), MAE (mean absolute error), and STD (standard deviation).

## Data Availability

Some of the content in this study is based on data, equipment, and engineering services provided by UNAVCO for the Plate Boundary Observatory operated by EarthScope (http://www.earthscope.org) (accessed on 26 March 2023). Soil moisture data were downloaded from the International Soil Moisture Network (https://ismn.geo.tuwien.ac.at/en/) (accessed on 26 March 2023).

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
