# Peer review of "Research on Soil Moisture Estimation of Multiple-Track-GNSS Dual-Frequency Combination Observations Considering the Detection and Correction of Phase Outliers"

_sensors, 2023, doi:10.3390/s23187944_

Round 1
Reviewer 1 Report
I think this manuscript has the following highlights:At the level of the dataset, the author constructed linear combinations of GNSS dual-frequency observations L4 and DFPC, respectively. Then, they computed the combined multipath errors, which can replace signal-to-noise ratio (SNR) for monitoring soil moisture changes at stations with one day for time resolution. At the analytical level, this work also evaluated the multivariate linear regression (MLR) and extreme learning machine (ELM) to construct multi-satellite linear regression models (MSLR) and multi-satellite nonlinear regression models (MSNR) for soil moisture prediction. There are a few comments on the article, and my advice is to revise it moderately.
1) In reference to line 121, the appearance of outliers in delayed phases was mentioned in the introduction, which leads to the detection and correction of outliers in the following section, and some literature on outlier handling should be introduced here to elaborate on the description.
2) Regarding line 506, doesn't clear why you limited the satellite altitude angle range to within 10° to 20°, with a detailed explanation in the response to the reviewer.
3) Regarding line 481, in order to ensure the uniqueness of the parameter solution, the number of combined multipath errors participating in the indirect leveling should not be smaller than the number of parameters to be solved, and please give some elaboration in the reply as to why this paper selects the combined multipath error sequence of length 5 to participate in the solving.
4) In the span of lines 598 to 606, soil moisture was predicted by linear model MLR and nonlinear model ELM, respectively, to compare and reflect the influence of environmental factors around the station on the prediction accuracy, and whether it is possible to analyze the influence of environmental factors on the prediction accuracy from a qualitative and quantitative point of view.
It is well-written, logically organized, and the figures and tables are appropriate.
Author Response
Dear Reviewer,
We would like to thank the anonymous reviewer for providing an opportunity to revise the manuscript. The comments and suggestions of the reviewer are all valuable and very helpful. We have studied them carefully and have made revisions to improve the manuscript. Revised portion are marked in red in the manuscript and the main corrections and additions have been uploaded as an attachment for the reviewer's consideration.
Best regards,
Authors

Reviewer 2 Report
Review Comments
This manuscript presents a fresh investigation of the moistures of soils based on the global navigation satellite system interferometric reflectometry (GNSS-IR) microwave remote sensing technology. The method consideres of the effect of indices involving moisture content, bulk density and fractal dimension of grain size, as well as their interactions. The interesting findings indicate a substantial achievement in high-precision soil 34 moisture estimation within a small satellite elevation angle range. Before possible publication of this article, following minor issues require to be addressed:
(1) Line 34: I think there is a missing period between “respectively” and “These”. There are many similar errors, please correct the author carefully.
(2) The review of the current study on residual soils is not comprehensive. The unique characteristic of GNSS-IR are not fully discussed in the Introduction section. A better outline and more updated citations are expected.
(3) The “soil moisture” only have two words. There is no need for using the abbreviation (SM).
(4) Figure 2 and 3 appear not very clear, some words are big, some are small, and the grouping lines are not quite clear. It needs to be amended.
(5) The contribution of considered indices and their interactions, which is examined and discussed in great detail in this study, are not noted in the Conclusions section. It is therefore need to be indicated.
There are a number of typos or non-standard expressions in the manuscript. In addition, a space should be normally provided between the value and the unit. Some spaces in the text are included and some are not. The authors need to read through the entire manuscript to check for corrections.
Author Response

(The authors gave the same response as above.)

Round 2
Reviewer 2 Report
I think the authors have made sufficient revisions to the manuscript. I recommand to accept the revised manuscript for publication.
The language is basically correted.